# Dynamic encoding of social threat and spatial context in the hypothalamus

Piotr Krzywkowski[1,2], Beatrice Penna[1,3], Cornelius T Gross[1]*

[1]Epigenetics & Neurobiology Unit, EMBL Rome, European Molecular Biology Laboratory, Monterotondo, Italy; [2]EMBL and Heidelberg University, Faculty of Biosciences, Heidelberg, Germany; [3]Masters Course in Biomedical Engineering, Faculty of Civil and Industrial Engineering, Sapienza University, Roma, Italy

**Abstract** Social aggression and avoidance are defensive behaviors expressed by territorial animals in a manner appropriate to spatial context and experience. The ventromedial hypothalamus controls both social aggression and avoidance, suggesting that it may encode a general internal state of threat modulated by space and experience. Here, we show that neurons in the mouse ventromedial hypothalamus are activated both by the presence of a social threat as well as by a chamber where social defeat previously occurred. Moreover, under conditions where the animal could move freely between a home and defeat chamber, firing activity emerged that predicted the animal's position, demonstrating the dynamic encoding of spatial context in the hypothalamus. Finally, we found that social defeat induced a functional reorganization of neural activity as optogenetic activation could elicit avoidance after, but not before social defeat. These findings reveal how the hypothalamus dynamically encodes spatial and sensory cues to drive social behaviors.

*For correspondence:
gross@embl.it

**Competing interests:** The authors declare that no competing interests exist.

## Introduction

Comparative molecular and functional studies across animal species demonstrate that the hypothalamus contains evolutionarily conserved brain networks for controlling survival behaviors and maintaining physiological homeostasis (*Swanson, 2000*; *Tosches and Arendt, 2013*). Work in laboratory rodents has shown that the medial hypothalamus, in particular, harbors distinct neural systems essential for defense and reproduction (*Canteras, 2002*). The best understood of the medial hypothalamic nuclei is the ventromedial nucleus (VMH) whose dorsal medial (VMHdm) and ventrolateral (VMHvl) divisions are key nodes in the defensive and reproductive systems, respectively (*Canteras et al., 1994*; *Canteras, 2002*). VMHdm is required for defensive responses to predators (*Silva et al., 2013*; *Kunwar et al., 2015*; *Viskaitis et al., 2017*), while VMHvl is necessary for mounting and territorial aggression, key reproductive behaviors (*Lin et al., 2011*; *Yang et al., 2013*). However, loss-of-function studies demonstrate that defensive responses to social threats do not depend on VMHdm, but rather require VMHvl, suggesting that the reproductive system plays a more general role in controlling both aggression and defense to social stimuli (*Silva et al., 2013*; *Silva et al., 2016a*). Consistent with a general role in social threat responding, electrical, pharmacogenetic, or optogenetic stimulation of VMHvl is able to elicit or increase the probability of social aggression (*Olivier, 1977*; *Kruk et al., 1983*; *Lin et al., 2011*; *Lee et al., 2014*; *Hashikawa et al., 2017*; *Yang et al., 2017*; *Wang et al., 2019*) and avoidance (*Sakurai et al., 2016*; *Wang et al., 2019*). However, these responses are often unreliable and have been shown to be influenced by the social and hormonal status of both the subject and the threat (*Lin et al., 2011*; *Lee et al., 2014*; *Sakurai et al., 2016*; *Yang et al., 2017*) suggesting a role for past experience or other environmental factors in dictating the behavioral output of VMHvl.

Neurons in VMHvl show firing patterns that correlate with social investigation and attack (*Lin et al., 2011*; *Falkner et al., 2014*; *Remedios et al., 2017*) and cFos and bulk calcium imaging approaches identified partially overlapping recruitment of neural activity during social aggression and defeat (*Motta et al., 2009*; *Sakurai et al., 2016*; *Wang et al., 2019*). However, single unit recordings have not been reported during social defeat or avoidance and it remains unclear what aspects of these behaviors are encoded in VMHvl and whether the overlap between aggression and defense reflects a common behavioral or internal state component. VMHvl receives major afferents from the medial amygdala that encodes information about conspecific identity (*Canteras et al., 1995*; *Swanson and Petrovich, 1998*; *Li et al., 2017*). However, VMH also receives inputs from lateral septum and subiculum that could convey contextual information (*Risold and Swanson, 1997*; *Silva et al., 2016b*; *Wong et al., 2016*; *Lo et al., 2019*) and both of these input pathways are able to modulate aggression (*Wong et al., 2016*; *Leroy et al., 2018*) suggesting that VMHvl is in a position to integrate sensory and spatial information to guide social behavior. Finally, ensemble neural activity elicited in VMHvl during social investigation can be reshaped by sexual experience, demonstrating a capacity for experience-dependent changes in VMHvl inputs (*Remedios et al., 2017*).

Here, we investigated the functional encoding of defense and aggression by VMHvl using single unit neural activity recording and optogenetic manipulation during male-male social encounters in laboratory mice. A majority of neurons in VMHvl were active in a way that was consistent with the encoding of a generalized internal state of threat, responding during a variety of social threat situations. Unexpectedly, following social defeat sets of neurons emerged that were activated by the context where the social threat had occurred or by the home cage where the animal resided, demonstrating the experience-dependent encoding of spatial context. Moreover, social defeat reshaped the neuron ensemble activity elicited during social interaction and optogenetic stimulation could elicit robust escape behavior in defeated animals, but not in undefeated controls. These findings demonstrate that VMHvl dynamically encodes social threat and spatial context states in a manner that can guide defensive behavior.

## Results

### Encoding of social threat

In order to better understand what aspects of defense are encoded in VMHvl neuron firing, we used in vivo microendoscopic calcium imaging to measure neuronal response properties in mice subjected to social defeat (*Figure 1a–e*). Mice were habituated to a home chamber from which they were given access to a corridor and far chamber for a brief period each day. On the social defeat day, mice were closed into the far chamber and an aggressive mouse was introduced. Following social defeat, the far chamber door was opened to allow the mouse to escape and exhibit approach-avoidance behavior toward the aggressor who remained restricted to the far chamber. Many neurons (100/246, 40%, Social+) showed a tonic increase in activity during the social defeat phase that returned to baseline levels.

Next, we examined neural activity patterns during the post-defeat, approach-avoidance phase. The mice repeatedly advanced in a cautious manner toward the aggressor, exhibiting frequent risk assessment behaviors in which they stretched their body in the direction of the far chamber (stretch-attend or stretch-approach, *Blanchard et al., 2011*). Once close to the far chamber mice often turned and fled back to the home chamber (flight, *Video 1*). Calcium imaging identified many cells that were robustly activated during risk assessment either close to or far away from the far chamber (160/326, 49%, Assessment+). Smaller sets of cells were either deactivated during risk assessment (63/326, 19%, Assessment-) or activated or deactivated during flight (30/198, 15%, Flight+; 60/198, 30%, Flight-; *Figure 1l–q*). Notably, Assessment+ cells were activated during risk assessment events that occurred at a distance from the far chamber, particularly at the junction of the home cage and corridor (*Figure 1o*; *Video 2*). A comparison of neuronal response properties revealed that a majority of Assessment+ cells (81/152, 53% vs chance 17%, p=$4.23\times10^{-11}$) overlapped with Defeat+ cells, suggesting that they may encode a generalized internal state associated with both direct threat as well as the assessment of threat even when this does not involve close social contact (*Figure 1r*). Moreover, many Assessment+ cells (37/100, 37% vs chance 14%, p=0.00031) were also Flight- cells, showing activation as the mouse approached the far chamber and then turning off abruptly when

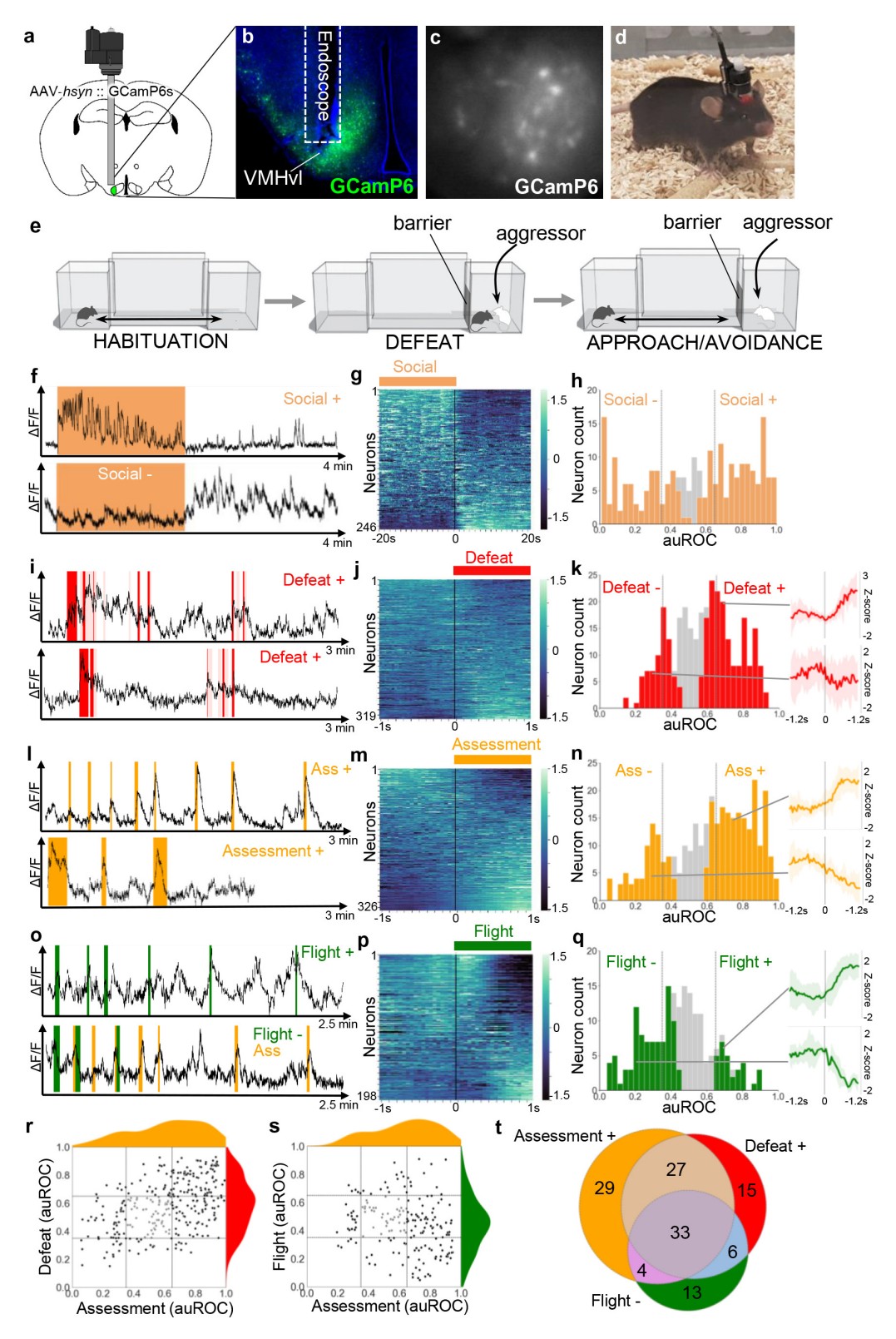

**Figure 1.** VMHvl encodes social threat. (**a–d**) Microendoscopy was used to image single unit calcium activity in VMHvl neurons of awake behaving mice expressing GCaMP6s. (**c**) Representative processed endoscope image used to extract relative changes in fluorescence (ΔF/F) for putative single neurons. (**e**, left) Mice were housed in a home chamber for several days during which time they were given access to a corridor and far chamber for 15 min each day. (**e**, middle) On the defeat day the mouse was enclosed into the far chamber and an aggressor was introduced who repeatedly attacked

*Figure 1 continued on next page*

*Figure 1 continued*

the mouse. (e, right) Subsequently, the barrier was opened and the defeated mouse exhibited approach and avoidance behavior toward the far chamber. (f, i, l, o) Activity traces of representative neurons showing increased (+) or decreased (-) signal during specific behaviors (color), light red in i represents mixed defense/upright behavior). (g, j, m, p) Summary of activity of all neurons across the onset or offset of each behavior. (h, k, n, q) Histogram of area under the receiver operator curve (auROC) for all neurons with significantly responding neurons indicated in color and average z-score ± SD (N = 5–12) traces of representative significant positive and negative responding neurons shown at right (p<0.05). Vertical lines in histogram indicate high (0.65) and low (0.35) cut-off for scoring positively responding neurons. (r, s) Correlation of neuron auROC scores across behaviors. Distributions are shown outside the axes. (t) Overlap of Defeat+, Assessment+, and Flight- neurons (N = 4) during the subsequent approach-avoidance phase (f–h). Other neurons (79/246, 32%, Social-) showed the opposite pattern, with unaltered or decreased activity during the social defeat phase and increased activity during the approach-avoidance phase (f–h). Although it appeared that Social- neurons might represent two distinct populations, one that decreased and another that remained unchanged during the social defeat phase, it was not possible to statistically distinguish them. A smaller set of neurons showed changes in activity that were time-locked to individual defeat events (117/319, 36%, Defeat+; 36/319, 11%, Defeat-) when the intruder attacked the experimental animal (i–k). These findings are consistent with earlier cFos and bulk calcium imaging studies (*Motta et al., 2009*; *Sakurai et al., 2016*; *Wang et al., 2019*) and confirm that VMHvl is strongly recruited during social defeat.

The online version of this article includes the following figure supplement(s) for figure 1:

**Figure supplement 1.** Complete behavioral paradigms.
**Figure supplement 2.** Spatial filter maps from all days of calcium imaging.

---

the animal fled (*Figure 1—figure supplement 1s*; *Video 1*). A similar firing response pattern has been reported for neurons in VMHdm during approach toward a predator (*Masferrer et al., 2018*). Overall, Defeat+, Assessment+, Flight- cells showed a high degree of overlap (26% vs chance 5%, p=2.0×10$^{-6}$) and made up the largest fraction of responsive neurons (*Figure 1t*). These findings suggest that VMHvl neurons may encode a generalized social threat state rather than particular behaviors associated with threat response.

## Dynamic encoding of spatial context

Our observation that a large fraction of neurons in VMHvl were activated during risk assessment both close and far from the social stimulus could be explained by the multi-sensory inputs that VMHvl receives (*Canteras et al., 1995*; *Garfield et al., 2014*; *Lo et al., 2019*; *Wong et al., 2016*). In particular, our earlier observation that VMHdm is required for the expression of defensive behaviors in a context previously associated with a predator suggested that the medial hypothalamus can be recruited by stimulus-associated cues (*Silva et al., 2016a*) and is consistent with evidence for activation of VMHvl during nose poke in anticipation of aggression (*Falkner et al., 2016*). To test for the recruitment of VMHvl by purely contextual cues, we performed in vivo calcium endoscopy in animals who experienced social defeat and were re-exposed 24 hr later to the defeat context in the absence of the aggressor (*Figure 2a*). Analysis of neuronal response properties revealed a large fraction of neurons with robust activation in the defeat chamber (93/343, 27%, Far chamber+, *Figure 2b–d*). Unexpectedly, a second group of neurons showed marked activation in the home chamber (78/343, 23%, Home+, *Figure 2e–g*). Notably, Home+ and Far chamber+ cells abruptly turned off when the animal passed from the chamber to the corridor, suggesting that they specifically responded to the far and home chamber contexts, rather than to a gradient of social threat cues, for example. A comparison of neuronal response properties between the context, approach-avoidance, and defeat phases of the test showed that Far chamber+ neurons strongly overlapped with Social+ neurons (45/67, 67% vs chance 11%, p=9.597×10$^{-11}$) and Home+ neurons with Social- neurons (37/60, 62% vs chance 7%, p=1.065×10$^{-10}$, *Figure 2i,j* and *Figure 2—figure supplement 1*) and that Far chamber+ neurons also overlapped with Defeat+ neurons (35/84, 42% vs chance 10%, p=2.0×10$^{-6}$; *Figure 2k*) and, in a three-way comparison, a larger fraction of Far chamber+ than Home+ neurons were Assessment+ or Defeat+ (*Figure 2l,o*). These

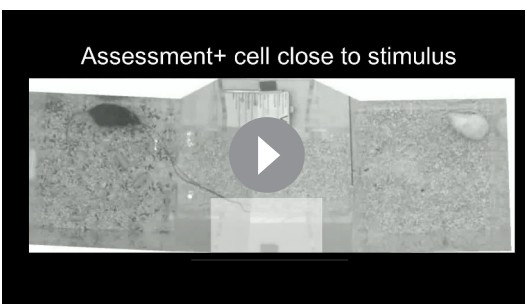

**Video 1.** Assessment+ cell close to stimulus.
https://elifesciences.org/articles/57148#video1

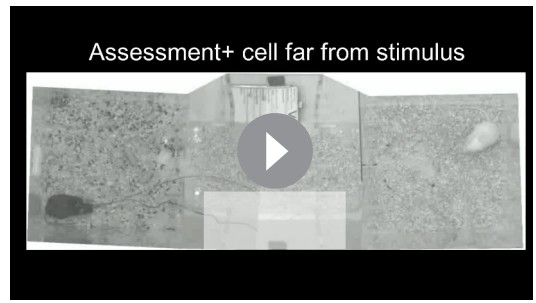

**Video 2.** Assessment+ cell far from stimulus.
https://elifesciences.org/articles/57148#video2

data suggest that VMHvl encodes a generalized state of social threat that can be reactivated by contextual cues (Social+, Defeat+, Far chamber +) as well as features of social territory (Home+).

To understand whether the recruitment by context was linked to the past experience of the animal, we compared context encoding in animals before and after social defeat. Only a small number of Far chamber+ and Home+ cells could be identified during the pre-defeat habituation phase even though the animal explored the apparatus to a similar extent in both phases (*Figure 2—figure supplement 2c*). To quantify changes in neuron activity before and after defeat, we employed linear discriminant analysis (LDA) of neuron ensemble activity. Before defeat, neuron ensemble activity in the home, corridor, and far chambers was overlapping as visualized using principal LDA discriminants (*Figure 2m,n* and *Figure 2—figure supplement 2a*). Following defeat, however, home, corridor, and far chamber ensemble activity became significantly more separated (*Figure 2p,q* and *Figure 2—figure supplement 2b*; Home vs. Far chamber LDA distance, p=0.0075) confirming the observation that social defeat induced a marked enhancement of the encoding of spatial context and suggesting that VMHvl may dynamically encode features of territory.

## Overlapping encoding of aggression and defense

Having established that VMHvl encodes features of social threat and context we examined the hypothesis that social threat neurons would also be active during aggression. First, we performed serial cFos labeling using TRAP-tagging (*cFos*::CreERT2; *Rosa26*::LSL-tomato; *Guenthner et al., 2013*) to determine the extent of overlap in recruitment during social defeat and resident-intruder aggression. Naive animals were subjected to two resident-intruder tests at one week interval in which they were either the resident or intruder, in all possible combinations (Aggression-Aggression, Defense-Defense, Aggression-Defense, Defense-Aggression). Immediately following the first test mice were treated with 4-OHT to induce persistent labeling of cFos+ cells, followed by immunolabeling of cFos+ cells after the second test. Animals subjected to the same behavioral experience (Aggression-Aggression or Defense-Defense) showed 50 ± 15% overlap of cFos+ cells while those with different experiences (Aggression-Defense or Defense-Aggression) showed 18 ± 5% overlap and those under control conditions 10 ± 2% overlap (ANOVA: F = 20.89, p=0.0001; *Figure 3a–d*). These data are consistent with experiments using a viral cFos-tagging strategy (*Sakurai et al., 2016*) and suggest that approximately one quarter of the cells in VMHvl activated during aggression and defense are recruited by both experiences and that this population of cells may support a common function during male-male social interaction.

Next, we used in vivo calcium endoscopy to understand whether VMHvl neurons recruited during aggression might overlap with the social threat and context neurons identified following defeat. Robust aggression was elicited by confining the mouse to its home cage and introducing a subordinate male mouse. During aggression phasic modulation of neuron activity occurred during bouts in which the resident actively investigated (56/310, 18% Sniff+; *Figure 3e–g*) or attacked (38/266, 14% Attack+; *Figure 3h–j*) the intruder. Comparison of neuronal response properties revealed that most Attack+ neurons (19/37, 51% vs chance 3%, p=2×10⁻⁶) were Sniff+ neurons (*Figure 3k*), consistent with a strong overlap between these populations in previous single unit recording studies (*Lin et al., 2011*; *Falkner et al., 2014*). Comparison of neuronal response properties across social defeat and aggression showed that a large fraction of Defeat+ neurons (24/68, 35% vs chance 6%, p=0.00003) were reactivated during active investigation of the subordinate (Sniff+), indicating that these may encode a common social threat state in both defenders and attackers (*Figure 3l*). A smaller fraction of Defeat+ neurons (12/60, 20% vs chance 5%, p=0.02458) were Attack+ neurons (*Figure 3m*) and a three-way comparison revealed that 42% of Defeat+ cells overlap with either Sniff+ or Attack+ (*Figure 3n*), a finding that is consistent with our cFos data (*Figure 3a–d*) and confirms the recruitment of both common and unique neuron ensembles during defense and aggression. Comparison

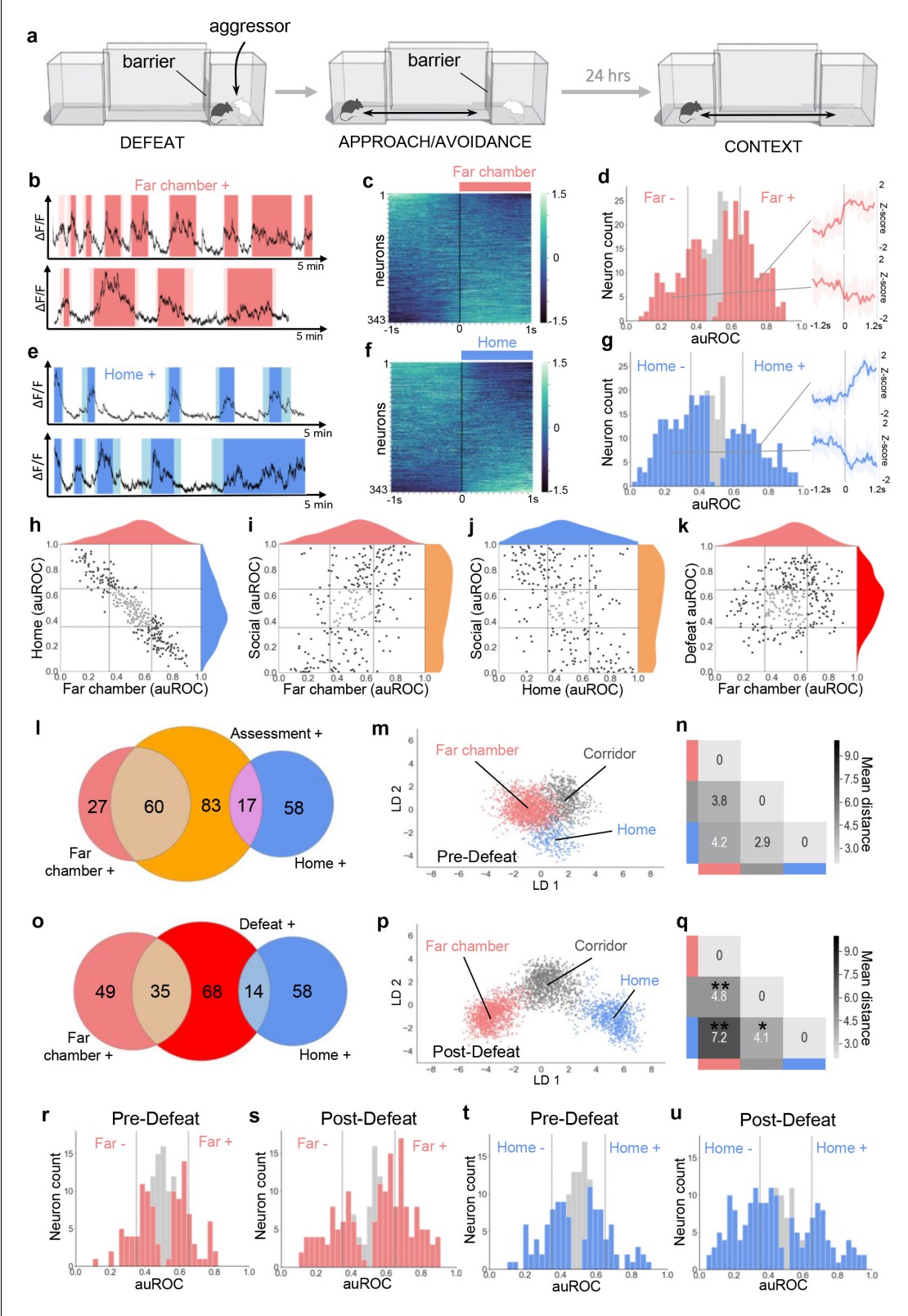

**Figure 2.** Dynamic encoding of spatial context in VMHvl. (**a**) In vivo calcium endoscopy was carried out in mice subjected to (left, middle) social defeat and then exposed again to (right) the defeat context one day later. (**b, e**) Activity traces of representative neurons showing increased (+) signal when the mouse enters the far or home chamber (dark color) or its immediately adjacent part of the corridor (light color). (**c, f**) Summary of activity of all neurons across the transition into the far or home chamber. (**d, g**) Histogram of area under the receiver operator curve for all neurons with significantly

*Figure 2 continued on next page*

*Figure 2 continued*

responding neurons indicated in color and average z-score traces ± SD (N = 9–10) of representative significant positive and negative responding neurons shown at right (p<0.05). Vertical lines in histogram indicate high (0.65) and low (0.35) cut-off for scoring positively responding neurons. (h–k) Correlation of neuron auROC scores between chambers and/or behavior. Distributions are shown outside the axes. (l, o) Overlap of Far chamber+, Home+, Assessment+, and Defeat+ neurons (N = 7). (m, p) LDA plot of neuron ensemble activity for a representative mouse in the home, far chamber, or corridor. Each data point represents a frame of calcium imaging data projected onto the first two linear discriminants. (n, q) Average distances between clusters of frames representing neuron ensemble activity for all mice (*p<0.05, **p<0.01, pre vs post - > F hr distance p=0.0075, t = 2.997, df = 9; F-C distance: p=0.005, t = 3.252, df = 9, H-C distance: p=0.037, t = 2.0213, df = 9). (r–u) Histograms of area under the receiver operator curve for all neurons with significantly responding neurons indicated in color. Vertical lines indicate high (0.65) and low (0.35) cut-off for scoring positively responding neurons. (m, n, r, t) Habituation phase before social defeat, and (p, q, s, u) context phase after social defeat (N = 4).

The online version of this article includes the following figure supplement(s) for figure 2:

**Figure supplement 1.** Overlap between social neurons.

**Figure supplement 2.** Representation of territory *before* and *after* defeat for all mice.

**Figure supplement 3.** Comparison of time spent in different chambers during memory day.

of aggression responsive neurons with those activated by the defeat context (Far chamber+) revealed an overlap between Attack+, Defeat+, and Far chamber+ neurons (*Figure 3o*). Unexpectedly, Attack+ neurons showed more overlap with Far chamber+ (14/34, 41% vs chance 4%, p=0.00022) than Home+ neurons (4/34, 12% vs chance 3%, p=0.35591; *Figure 3p*) despite the fact that the attack occurred in the home cage, reinforcing the idea that aggression elicits features of social threat, rather than safety.

## Social defeat remodels neural activity

Following on our observation that social defeat elicited changes in contextual encoding in VMHvl (*Figure 2*) and previous experiments showing that sexual experience can remodel VMHvl ensemble activity elicited by exposure to a social stimulus (*Remedios et al., 2017*), we investigated the possibility that social defeat might also induce changes in neural activity elicited during social encounter. Animals were exposed to repeated social defeat experiences in the far chamber of a dual chamber apparatus as described earlier (*Figure 1e* and *Figure 1—figure supplement 1*). The experimental animals showed similar frequencies of behaviors across the two days of social defeat (*Figure 4—figure supplement 1*). Nevertheless, to minimize any confounding effect of potential changes in behavior across experimental sessions, we restricted our ensemble analysis to data extracted from bouts of close social encounter (defeat, upright, orientating, sniffing, following, see Materials and methods). Linear discriminant analysis (LDA) was used to track and quantify shifts in neuron ensemble encoding across sessions. Neuron ensemble activity elicited during the first and second social defeat experiences were significantly non-overlapping (*Figure 4ab* and *Figure 4—figure supplement 2a*). As a control, we used LDA to track and quantify neuron ensemble activity across sessions of resident-intruder aggression carried out as previously described (*Figure 4—figure supplement 2c*). Unlike social defeat and consistent with previous studies (*Remedios et al., 2017*) repeated aggression was not associated with changes in neuron ensemble activity (*Figure 4ab* and *Figure 4—figure supplement 2a*) demonstrating that defeat has a unique capacity to alter the encoding of social cues in VMHvl. Notably, this change was found only when defeats preceded, but not when they followed aggression experiences (*Figure 4—figure supplement 2bc*). Notably, the changes in ensemble activity across defeat days were not associated with a change in defensive behaviors, although, as expected, upright postures were significantly lower in the group that experienced aggression first (*Figure 4—figure supplement 1*). This finding demonstrates that changes in neuron ensemble encoding seen across days are likely due to experience-dependent plasticity rather than spontaneous fluctuations in firing patterns, and suggests that repeated exposure to aggression and winning may impart resistance to the transforming effects of social defeat on social encoding in VMHvl.

## Dynamic encoding of social threat by Esr1+ neurons

A previous in vivo calcium endoscopy study found activation of estrogen receptor α expressing (Esr1+) neurons in VMHvl during social interaction (*Remedios et al., 2017*) and a recent bulk calcium imaging study (*Wang et al., 2019*) as well as single-cell RNA-Seq study (*Kim et al., 2019*) reported activation of Esr1+ neurons during both social aggression and defeat suggesting that this

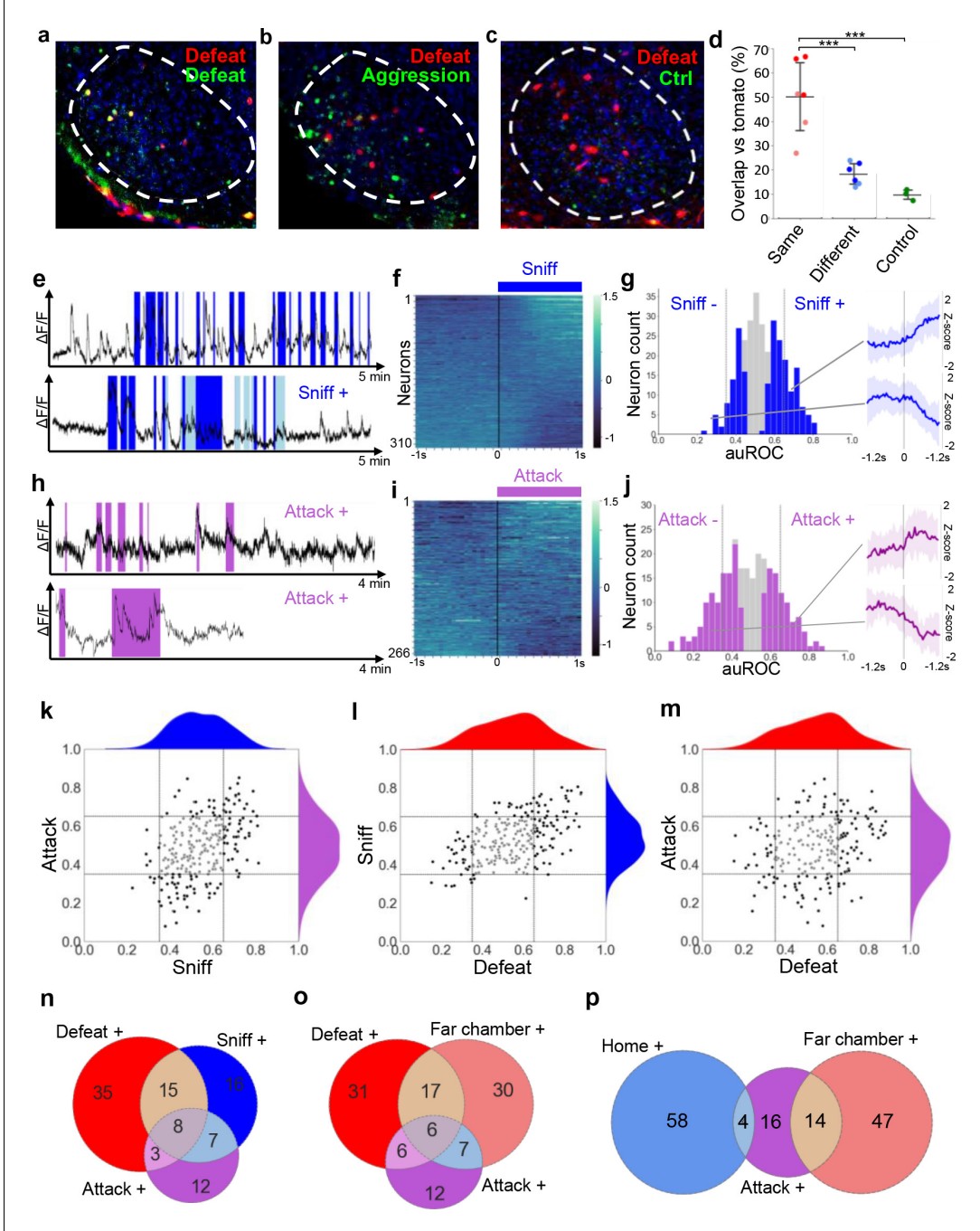

**Figure 3.** Overlapping encoding of social defense and aggression. (**a–c**) Representative images of cFos-tagged (red) and cFos-immunolabeled (green), and double labelled (yellow) cells in brain sections from mice subjected sequentially to social defeat and aggression in a counterbalanced manner. (**d**) Average percentage overlap of defeat and aggression recruited cFos+ cells as revealed by the difference between cFos+ overlap for same, different, or control behaviors (dark red: Defeat-Defeat, dark blue: Defeat-Aggression, green: Defeat-Control; light red: Aggression-Aggression, light blue: Aggression-Defeat; Same vs. Different, p=0.0005 N=6; Same vs. Control, p=0.0004, N = 3–6; Different vs. Control, p=0.4969, N = 3–6, F = 20.889; bars represent SD). (**e, h**) Activity traces of representative neurons showing increased (+) signal during close social investigation (Sniff+, dark blue; ano-genital sniffing, light blue) and aggression (Attack+). (**f, i**) Summary of activity of all neurons across the onset of behavior. (**g, j**) Histogram of area under the receiver operator curve for all neurons with significantly responding neurons indicated in color and average z-score traces ± SD (N = 22–34) of representative significant positive and negative responding neurons shown at right (p<0.05). Vertical lines in histogram indicate high (0.65) and low (0.35) cut-off for scoring positively responding neurons. (**k–m**) Correlation of neuron auROC scores among aggression behaviors and between aggression and defense behaviors. Distributions are shown outside the axes. (**n–p**) Overlap of aggression, defense, and territory-related neurons (N = 5).

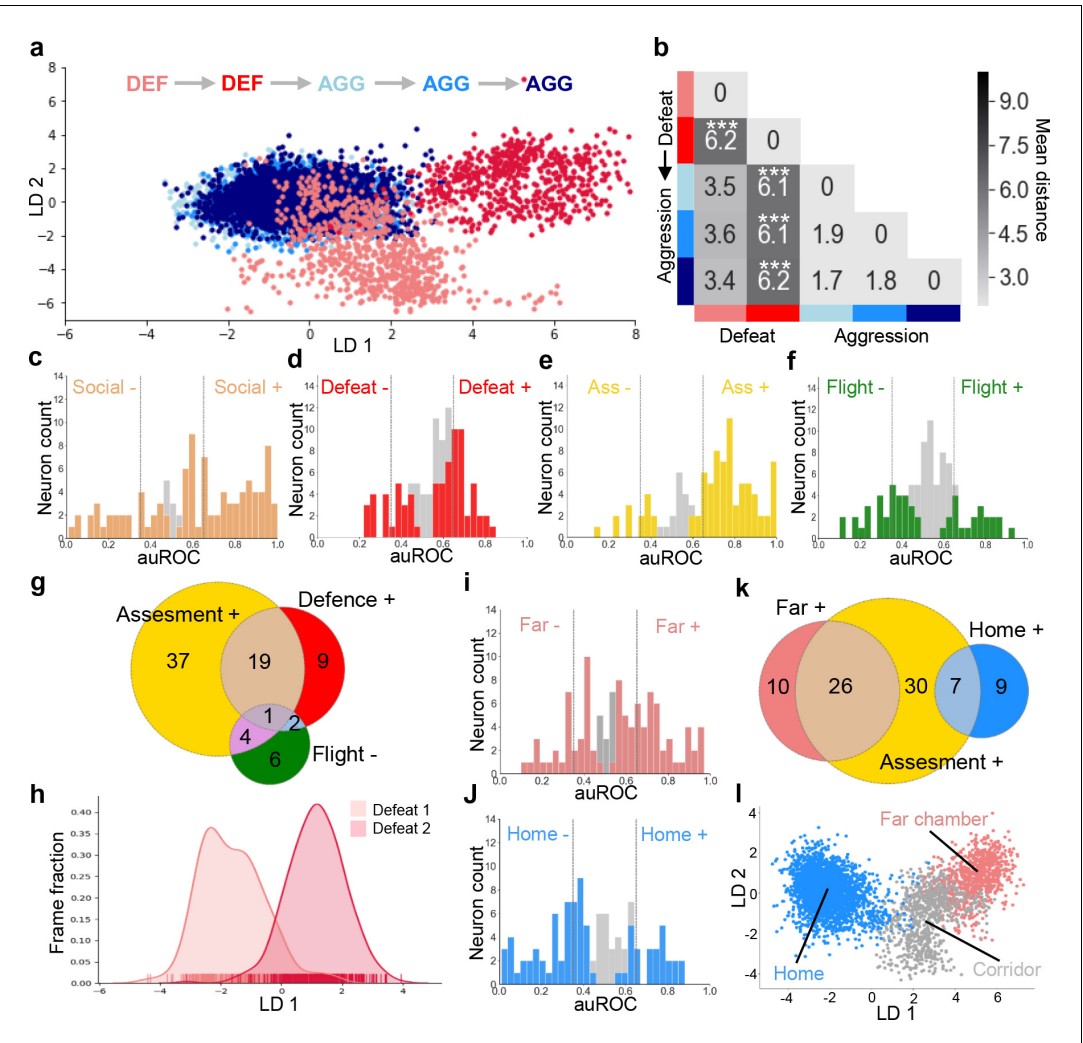

**Figure 4.** Social defeat remodels VMHvl activity. (a) LDA plot of neuron ensemble activity for a representative mouse during repeated defeat and aggression episodes. Each data point represents a frame of calcium imaging data projected onto the first two linear discriminants. (b) Average distances between clusters of neuron ensemble data between defeat and aggression episodes for all mice during forward order testing (N = 4; colors refer to episodes in a; *p<0.05, **p<0.01, ***p<0.001; distance between D1 and D2 vs distances between aggression: p=0.0002; distance between D2 vs aggressions p=0.0001; distance between D1 vs aggressions p=0.055; ANOVA F = 16.93, p=0.0001). Next, microendoscopy was used to image single unit calcium activity in VMHvl Esr1+ neurons of awake behaving mice expressing GCaMP6s during social defeat and post-defeat avoidance as done before in the general population of VMHvl neurons. (c–f) Histogram of area under the receiver operator curve (auROC) for all Esr1+ neurons with significantly responding units indicated in color. Vertical lines in histogram indicate high (0.65) and low (0.35) cut-off for scoring positively responding neurons. (g) Overlap of Defeat+, Assessment+, and Flight- neurons (N = 78). Finally, in vivo calcium endoscopy was carried out in Esr1+ neurons of mice subjected to social defeat and then exposed again to the defeat context one day later. (h–i) Histogram of area under the receiver operator curve for all Esr1+ neurons with units showing significant responding to the (h) far or (i) home chamber indicated in color. Vertical lines in histogram indicate high (0.65) and low (0.35) cut-off for scoring positively responding neurons. (j) Overlap of Assessment+, Far chamber+, and Home- neurons (N = 82). (k) LDA plot of Esr1+ neuron ensemble activity for a representative mouse in the home, far chamber, or corridor. Each data point represents a frame of calcium imaging data projected onto the first two linear discriminants. (l) LDA plot of Esr1+ neuron ensemble activity for a representative mouse during sequential defeat episodes. Each data point represents a frame of calcium imaging data projected onto the first linear discriminant.

The online version of this article includes the following figure supplement(s) for figure 4:

**Figure supplement 1.** Comparison of behavior between days and groups.

**Figure supplement 2.** Representation of defeat and aggression states in LD space for all mice.

**Figure supplement 3.** Representation of defeat day 1 and defeat day 2 in LD space for all mice and esr1+ mice.

subpopulation of VMHvl neurons may contribute to the encoding of social threat we observed in the general population of VMHvl neurons. To test this hypothesis, we carried out in vivo calcium endoscopy in mice expressing GCaMP6 selectively in VMHvl Esr1+ neurons. Single unit neural activity was assessed from mice during social defeat, post-defeat approach/avoidance, and during re-exposure to the defeat chamber as described earlier (*Figure 2a*). During the social defeat phase of the test, many Esr1+ neurons showed tonic increases (47/102, 46%, Social+) or decreases (18/102, 18%, Social-) that returned to baseline levels during the subsequent approach-avoidance phase (*Figure 4c*). A subset of Esr1+ neurons showed phasic responses that were time-locked to individual defeat events (32/102, 31%, Defeat+; 11/102,11%, Defeat-; *Figure 4d*). During the subsequent approach/avoidance phase, many Esr1+ cells were robustly activated (63/98, 64%, Assessment+) or deactivated (7/98, 7%, Assessment-) during risk assessment and flight behavior (16/98, 16%, Flight+; 15/98, 15%, Flight-; *Figure 4e–f*). Overall, Defeat+ and Assessment+ cells showed a moderate, but not significant degree of overlap (32% vs chance 20%, p=0.44031) and made up a major fraction of responsive neurons (*Figure 4g*). These findings demonstrate that VMHvl Esr1+ neurons encode a generalized social threat state in a manner largely similar to the general VMHvl neuron population.

Analysis of neuronal response properties during context re-exposure revealed a large fraction of neurons with robust activation in the far chamber (37/106, 35%, Far chamber+, *Figure 4h*) or home (19/106, 18%, Home+, *Figure 4i*) chambers. A comparison of neuronal response properties between the context and approach-avoidance phases of the test showed that a larger fraction of Far chamber + than Home+ neurons were Assessment+ (*Figure 4j*) and linear discriminant analysis of ensemble neural encoding in Esr1+ cells showed a clear separation of firing patterns in Home and Far chambers (*Figure 4k*). These data confirm that VMHvl Esr+ neurons encode features of social territory in a similar way to the general VMHvl population.

Finally, to understand whether social defeat could remodel neuronal ensemble activity in VMHvl Esr1+ neurons animals were exposed to two consecutive rounds of social defeat as described earlier (*Figure 1e*). Linear discriminant analysis (LDA) of neuron ensemble activity extracted from bouts of close social encounter (defeat, upright, orientating, sniffing, following, see Materials and methods) revealed that neuron ensemble activity elicited during the first and second social defeat experiences were significantly non-overlapping (*Figure 4l* and *Figure 4—figure supplement 3c*; Kolmogorov-Smirnov test, p=0.0001 for each mouse). Machine learning using Esr1+ cells data achieved similar levels of accuracy in distinguishing defeat days as using the data from all cells (*Figure 4—figure supplement 3d*). These data confirm that Esr1+ ensemble activity patterns are remodeled by social defeat in a manner indistinguishable from the general VMHvl population.

## Functional remodeling of Esr1+ neurons by social defeat

Changes in neuron ensemble encoding in VMHvl following social defeat could reflect plasticity in upstream brain regions that provide afferent inputs to VMHvl or they could be due to plasticity within VMHvl. These possibilities can be distinguished using a gain-of-function approach. Artificial activation of VMHvl before and after social defeat should result in different behavioral outputs if plasticity occurs within or downstream of VMHvl, but not if it is restricted to afferent inputs. Previous work showed that optogenetic activation of VMHvl Esr1+ neurons can elicit aggression against females or castrated males (*Lee et al., 2014*; *Wang et al., 2019*). However, in some cases, such animals showed brief avoidance responses during the initial stimulation trials (*Lee et al., 2014*) suggesting that VMHvl Esr1+ neurons are capable of promoting both aggression and avoidance, but that this may occur in in a manner that depends on context or past social experience.

To test whether social defeat might induce changes in VMHvl Esr1+ function, we expressed the blue light-activated cationic channel, channelrhodopsin 2 (ChR2), in Esr1+ VMHvl neurons (AAV-*Ef1a*::FLEX-ChR2, *Esr1*::Cre mice) and subjected the animals to two sessions of social defeat (*Figure 5a–c*). Several hours after social defeat, a subordinate intruder mouse was introduced into the home cage of the animal and following a period of habituation, light pulses were delivered to activate Esr1+ neurons. Following social defeat optogenetic activation of Esr1+ neurons elicited rapid and robust defensive behaviors, including freezing, flight, a reduction in social interaction, and immobility in all animals as well as in a subset of mice (N = 3) that had not been previously stimulated (*Figure 5d*). In one particularly striking case, optogenetic activation of Esr1+ neurons occurring while the resident was attacking the intruder caused an abrupt cessation of aggression and evoked sudden flights away from the subordinate animal (*Video 3*). Notably, no significant increase in

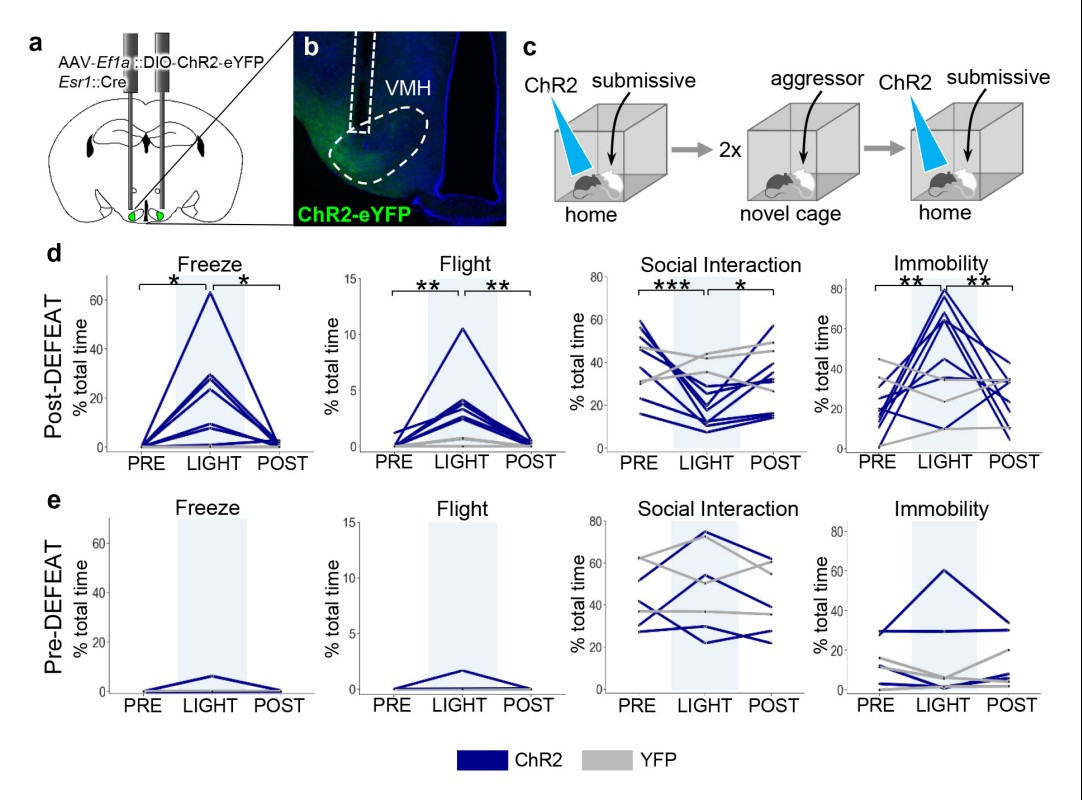

**Figure 5.** Social defeat remodels VMHvl function. (**a–b**) Optogenetic stimulation of Esr1+ neurons following local delivery of AAV-*Ef1a*::FLEX-ChR2-eYFP into the VMHvl of *Esr1*::Cre mice. (**c**) Mice were stimulated intermittently (20 Hz, 20 ms pulse, 30 s ON) following the introduction of a subordinate mouse into the home cage before (Pre-Defeat) and/or after (Post-Defeat) two episodes of social defeat in the far chamber. (**d–e**) Trial-averaged (N = 4–5 trials) behavioral measures before (Pre, 30 s), during (Light, 30 s), and after (Post, 30 s) optogenetic stimulation of Esr1+ neurons in VMHvl during either the (Blue – mice expressing ChR2; Grey – control mice expressing YFP) (**d**) Post-Defeat or (**e**) Pre-Defeat episodes (N = 3–7, *p<0.05, **p<0.01, ***p<0.001; Pre-defeat: Freezing – ANOVA F = 1, p=0.42, Flight – ANOVA F = 1, p=0.42, Interaction – ANOVA F = 0.68, p=0.54, Immobility – ANOVA F = 0.25, p=0.79; Post-defeat: Freezing – ANOVA F = 7.1, p=0.0075, Tukey's: Pre vs Light p=0.013, Light vs Post p=0.017, Pre vs Post p=0.99, Flight – ANOVA F = 13.8, p=0.0005, Tukey's: Pre vs Light p=0.0012, Light vs Post p=0.0012, Pre vs Post p=0.10, Interaction – ANOVA F = 19.15, p=0.0001, Tukey's: Pre vs Light p=0.0001, Light vs Post p=0.019, Pre vs Post p=0.021, Immobility – ANOVA F = 11.2, p=0.0012, Tukey's: Pre vs Light p=0.0014, Light vs Post p=0.0083, Pre vs Post p=0.63).

defensive behavior was elicited by optogenetic activation in ChR2-expressing mice that were stimulated during a resident-intruder test on the day before the social defeat (*Figure 5e*), nor in mice expressing a control virus (AAV-*Ef1a*::FLEX-YFP, *Esr1*::Cre). These data demonstrate that VMHvl neurons promote defense in a manner that depends on social experience and argue that at least part of the plasticity in neuron ensemble recruitment during social interaction (*Figure 4ab*) occurs within or downstream of VMHvl.

## Discussion

By performing single unit in vivo recordings during social defeat as well as during the post-defeat approach-avoidance period, again during re-exposure to the threat context one day later, and during resident-intruder aggression we were able to examine the encoding of social threat in the VMHvl across a wide variety of defensive behaviors. Neurons activated when the animal was attacked were reactivated later when the animal performed risk assessment behaviors (*Figure 1l,o*; *Video 2*) and many of them were reactivated when the animal explored the chamber where the defeat occurred one day later (*Figure 2l,o*). Moreover, a significant fraction of social threat cells were activated when

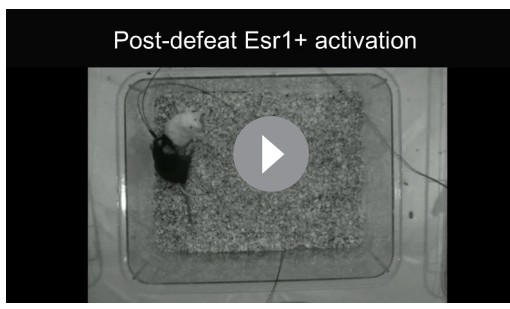

**Video 3.** Post-defeat optogenetic activation of Esr1+ neurons in VMHvl.
https://elifesciences.org/articles/57148#video3

the same mouse sniffed or attacked a subordinate mouse in its home cage (resident-intruder test, *Figure 3n*) consistent with the view that conspecific aggression in mice is driven by a defensive instinct aimed at expelling competitors from their territory and ensuring access to reproductive and nutritional resources (*Lorenz, 1963*). Nevertheless, overall, neurons were more robustly activated when the animal was being attacked than when they were attacking, suggesting a more efficient recruitment of VMHvl by threat in the losing animal (*Figure 1i–n* and *Figure 3h–j,n*). These data allow us to conclude that a major function of VMHvl is to provide an internal state of social threat that can be generalized across a wide variety of defensive behaviors and cues.

Our observation that social defeat drove the rapid emergence of a unique set of cells that fired in the home chamber and did not overlap with those encoding social threat (*Figure 2h–i,o–q*) was unexpected and shows that VMHvl encodes aspects of spatial context beyond those associated with threat. Although the function of these home cells is presently unknown, we speculate that they encode aspects of the animal's territory and that they may function to support territorial behaviors such as scent marking, sex, and defensive aggression. We noted that Home+ and Far chamber+ cells were modulated abruptly when the animal entered or exited the relevant chamber, suggesting that they encode context rather than a gradient of threat (*Figure 2b,e*). VMHvl receives inputs from the ventral hippocampus via LS that could contribute territory-related contextual information (*Risold and Swanson, 1997*; *Strange et al., 2014*) and are able to modulate conspecific aggression in VMHvl (*Leroy et al., 2018*; *Wong et al., 2016*). Our findings open up the possibility that contextual and social cues are integrated in a dynamic manner in VMHvl to drive and modulate a wide variety of behaviors aimed at territorial defense.

Our neuron ensemble analysis shows that social defeat reshaped both spatial context as well as social cue encoding. Ensemble neuron activity elicited during close social investigation became transformed following a social defeat experience, but not following a social aggression experience (*Figure 4a–b*). The latter observation is consistent with a previous recording study in which resident-intruder aggression failed to induce changes in VMHvl ensemble activity, while sex did (*Remedios et al., 2017*). Finally, our gain-of-function experiment demonstrates that the social defeat-induced transformation of encoding involves functional plasticity in or downstream of VMHvl Esr1+ neurons (*Figure 5c–d*). We note that in our hands optogenetic stimulation of VMHvl Esr1+ neurons was unable to elicit reliable aggression behavior against intact male intruders under baseline conditions, a finding that is consistent with previous studies where such stimulation only elicited reliably aggression against females or castrated males and in which a significant fraction of animals showed brief avoidance behavior upon initial optogenetic stimulation (*Lee et al., 2014*; *Lin et al., 2011*). We interpret these findings to indicate that VMHvl is primed to promote either defensive avoidance or defensive aggression under baseline conditions in a manner that depends on past social experience. This interpretation is supported by a study showing that pharmacogenetic activation of progesterone receptor expressing neurons in VMHvl can increase the reliability of attack in the resident-intruder assay if the animal was previously singly housed, but not if it was group housed (*Yang et al., 2017*).

Our observations reveal the encoding and control of instinctive motivational states by VMHvl neurons, showing that these encode an experience-dependent map of spatial context and that they exhibit functional plasticity in response to past social experience that guides the selection of instinctive behavioral outputs. These data argue for a reevaluation of the role of the hypothalamus in behavior. Rather than being viewed as a hardwired, innate behavioral response region, it should be seen as an integrator of present and past sensory and contextual information that adapts survival behaviors to a changing environment.

# Materials and methods

## Key resources table

| Reagent type (species) or resource | Designation | Source or reference | Identifiers | Additional information |
|---|---|---|---|---|
| Strain, strain background (*M. musculus*) | C57BL/6J | Charles River | RRID:IMSR_JAX:000664 | |
| Strain, strain background (*M. musculus*) | CD-1 | Charles River | RRID:IMSR_CRL:022 | |
| Strain, strain background (*M. musculus*) | BALB/c | EMBL | | Internal mouse colony at EMBL Rome |
| Genetic reagent (*M. musculus*) male | Esr1:Cre | Jackson Laboratory | RRID:IMSR_JAX:017913 | |
| Genetic reagent (*M. musculus*) male | *cFos*::CreERT2 | Jackson Laboratory | RRID:IMSR_JAX:021882 | |
| Genetic reagent (*M. musculus*) male | *RC*::LSL-tdTomato | Jackson Laboratory | RRID:IMSR_JAX:007914 | |
| Antibody | goat polyclonal anti-cFos SC-52G | Santa Cruz | RRID:AB_2629503 | |
| Chemical compound, drug | 4-hydroxytamoxifen 70% z-isomer | Sigma | H6278 CAS: 68392-35-8 | |
| Strain, strain background (*M. musculus*) | AAV-EF1a-DIO-hChR2 (E123T/T159C)-EYFP | UNC Vector Core | | |
| Strain, strain background (*M. musculus*) | AAV-EF1a-DIO-EYFP | UNC Vector Core | | |
| Strain, strain background (*M. musculus*) | AAV5-*CAG*::Flex-GCaMP6s | Penn Vector Core | | |
| Strain, strain background (*M. musculus*) | AAV5-*hSyn*::GCaMP6s | Penn Vector Core | | |
| Software, algorithm | ImageJ | https://imagej.nih.gov/ij/ | RRID:SCR_003070 | |
| Software, algorithm | TurboReg | http://bigwww.epfl.ch/thevenaz/turboreg/ | RRID:SCR_014308 | *Thevenaz et al., 1998* |
| Software, algorithm | Observer XT11 | Noldus | | https://www.noldus.com/observer-xt-animal |
| Software, algorithm | Prism 5 | Graphad | RRID:SCR_002798 | https://www.graphpad.com |
| Software, algorithm | V2.2 Radiant | Plexon | | https://plexon.com/products/plexbright-4-channel-optogenetic-controller-radiant-v2/ |
| Software, algorithm | Salomon Coder | https://solomon.andraspeter.com/ | | Behaviour coding software |
| Software, algorithm | scikit-learn | https://scikit-learn.org | RRID:SCR_002577 | *Pedregosa et al., 2012* |

## Animals and behavioral apparatus

All experimental procedures involving the use of animals were carried out in accordance with EU Directive 2010/63/EU and under approval of the EMBL Animal Use Committee and Italian Ministry of Health License 541/2015-PR to C.G. Animals were maintained in a temperature and humidity controlled environment with food and water provided ad libitum and 12 hr/12 hr light-dark cycle with lights on at 7:00. Experimental male C57BL/6J wild type (Charles River) and *Esr1*::Cre (Stock No.

017913, Jackson Laboratory) mice were switched to reverse dark-light cycle (lights off at 9:00) and singly housed in the home cage of the behavioral apparatus at least 1 week before initiating the experimental procedures and tested during the dark period under red lighting (two 1W LED lights). The custom Plexiglas behavioral apparatus consisted of three detachable parts: 1) home cage with dimensions 25 × 25×25 cm with a Y-shaped slit, 4 cm wide at the bottom serving as an entrance closed by sliding doors, 2) stimulus chamber identical to home cage, and 3) a corridor 46 × 12×30 cm connecting home cage and stimulus chamber (modified from *Silva et al., 2013*). Aggressor mice were CD-1 adult retired breeders (Charles River) screened for robust aggression and singly housed (*Franklin et al., 2017*). Subordinate mice were 9–15 week old BALB/c males bred at EMBL. Mice cages were changed weekly.

## Surgical procedures

Mice were anesthetised before surgery with 3% isoflurane (Provet) in oxygen and placed in stereo-taxic frame (Kopf). Anesthesia was maintained with continuous 1–2% isoflurane administration in breathing air enriched with oxygen. Body temperature was maintained with a heating pad. During surgery the skull was exposed, aligned, and cleaned with 0.3% hydrogen peroxide solution. For optogenetic activation experiments, 0.1–0.2 µl of AAV5-*Ef1a*::DIO-hChR2(E123T/T159C)-EYFP or AAV5-*Ef1a*::DIO-EYFP virus (UNC Vector Core) was infused bilaterally into VMHvl (from Bregma L:+ / - 0.67, A/P: −0.98, D/V: −5,75). After 5–10 min, the glass capillary was retracted and custom-made optic fibre connectors were implanted (0.66 NA, 200 µm core fibre and ceramic ferule with 230 µm/1250 µm internal/external diameter; from Bregma L: + / - 0.67, A/P: −0.98, D/V: −5.55 and L: + / - 1.14, A/P: −0.98, D/V: −5.6, at 5° angle). For in vivo calcium endoscopy 0.2–0.3 µl of AAV5-*hSyn*::GCaMP6s or AAV5-*CAG*::Flex-GCaMP6s (Penn Vector Core) virus was injected unilaterally into VMHvl and the endoscope lens (Snap-imaging cannula model L type E, Doric Lenses) was implanted at a very slow rate (from Bregma L: + / - 0.67, A/P: −0.98, D/V: −5.7). All implants were secured to the skull using miniature screws (RWD) and dental cement (Duralay). The wound was cleaned and skin was stitched around the implant. After the surgery mice received intra-peritoneal injection of 0.4 ml saline and were placed into heated cages with drinking water containing paracetamol for ~1 week. Mice were maintained in isolation for ~4 weeks before experimentation.

## Social defeat test

On the social defeat day, the mouse was allowed to explore the apparatus freely for 5 min, after which the animal was closed in the stimulus chamber and an aggressor was introduced for 10 min. In a few cases, the animal was allowed to escape earlier to avoid excessive defeat. The mouse was released from the stimulus chamber and the door closed to confine the aggressor to the stimulus chamber. After social defeat the mouse was allowed to explore the apparatus freely for at least 5 min. The memory test was conducted on the day following after social defeat, during which the mouse could explore the apparatus freely. The apparatus was washed with detergent and 50% alcohol between mice to avoid any remaining smell that could influence behavior of the test subject. CD-1 aggressors were screened for aggression for 3 days. Every day an intruder was placed in the aggressor's cage for 3 min. Only mice that attacked the intruder on every occasion were selected.

## Aggression test

On the aggression day a BALB/c intruder was introduced for 10 min after which the intruder was removed. BALB/c intruders used for this test were 9–15 weeks old and housed 3–5 mice per cage. For cFos experiments, animal were allowed to explore the entire apparatus for 5 min before and after introduction of the intruder in order to match exploration in the social defeat test.

## Behavioral data acquisition and annotation

Behavior was recorded at 40–50 Hz from above with up to two cameras (acA1300-60gmHIR, Basler) with GigE connections using Pylon software. Frame-by-frame behavioral annotation was carried out manually using Observer XT11 (Noldus) and Solomon Coder software. The experimenter was blind to genotype or calcium trace of recorded neurons when scoring behavior. The following behaviors were scored: defeat – biting attack toward the experimental animal in which the animal exhibited avoidance behavior (whole body movement away from intruder); *assessment* – stretch attend or

stretch approach behavior in which the animal extended its body in the direction of the threat or threat chamber from an immobile or slowly moving position; *flight* – rapid movement away from the threat or threat chamber; attack – biting attack or vigorous anogenital sniffing toward the intruder animal; *sniffing* – close contact of the nose of the experimental animal with the intruder; upright – animal rises on back paws with head up, keeping front paws stretch out; put down – keeping other animal down usually with two front paws and staying on top of it; follow – experimental animal closely follows intruder; cornering – staying in the corner of the apparatus (animal has body contact with two walls); locomotion – free movement and exploration of experimental apparatus; freeze – no body movement; head/body orientation – turning head or whole body towards another conspecific. Any animals with misplaced viral infections or optic fiber implants were excluded from the analysis.

## Histology

All animals were deeply anesthetized and transcardially perfused with PBS (Invitrogen) followed by 4% PFA (Sigma) in 0.1M PB solution and then post-fixed in 4% PFA at 4°C for 24 hr and cut using a vibratome (Leica VT 1000 s) in PBS (80 µm) for injection or implant location check, or staining (50 µm) procedure. If not used immediately, sections were stored in PBS with 0.1% sodium azide. For anti-cFos staining (SC-52G, Lot FO215, Santa Cruz) sections were washed three times in PBS for 10 min, blocked with 10% normal donkey serum, 0.2% Triton-X in PBS for at least 1 hr, and incubated with primary antibody (1:500) containing 5% normal donkey serum, 0.2% Triton-X overnight at 4°C. Sections were washed three times in PBS, incubated with secondary antibody solution (1:1000) containing 10% normal donkey serum for 2 hr at room temperature, washed twice with PBS, and stained with DAPI for 15 min, before washing twice with PBS and mounting on SuperFrost Plus slides (ThermoFischer) with Moviol.

## cFos mapping

Double heterozygous *cFos*::CreERT2;*RC*::LSL-tdTomato mice (Stock No. 021882 and 007914, Jackson Laboratory) were isolated for 7 days and subjected to 3 days of habituation to the apparatus before social defeat. A single dose (50 mg/kg) of 4-hydroxytamoxifen (4-OHT, 70% z-isomer, Sigma) was injected intra-peritoneally <2 min after social defeat to induce fluorophore expression in neurons recruited by defeat. One week later, mice were again habituated for 3 days to the apparatus and subjected to social aggression, and 1.5 hr later trans-cardially perfused and processed for cFos immunofluorescence. In a second group, the order of social defeat and social aggression were swapped to produce five experimental groups: defense-defense (same), aggression-aggression (same), defense-aggression (different), aggression-defense (different), defense-control (control). Control indicates mice exposed only to context prior to labelling.

## Optogenetic experiments

Heterozygous *Esr1*::Cre mice were injected with virus and allowed to recover for 2–3 weeks, housed under reverse light-cycle for 2 weeks, and then handled and habituated to the optical cables for at least 2 days prior to testing. Initially, after a 2-min free exploration period, each animal was stimulated with light (30 s, 20 Hz) every 1–2 min with increasing power (0.5, 1, 3, 6, 10 mW) to identify the optimal intensity to elicit a behavioral response (immobility or locomotion) and this was subsequently used for further stimulation (3 mW for most animals, with some responding at 1 mW and 6 mW). Control animals received 6 mW stimulation. For stimulation in the presence of a subordinate intruder-free exploration was allowed for ~3 min during which a single light stimulus was delivered (30 s, 20 Hz) after which a BALB/c intruder was introduced and the animal was further stimulated (3– 5 times, 30 s, 20 Hz) during periods of social interaction. Mice were then subjected to 2 days of social defeat (5 min) by a CD-1intruder in a novel plexiglas cage following 1 min free exploration. Several hours later, stimulation in the presence of a subordinate intruder was repeated as above. To control for possible repeated optogenetic activation effects a subset of animals from the ChR2 group (N = 3) was not stimulated before defeat episodes. Optical stimulation of ChR2 was achieved using a 465 nm LED (Plexbright, Plexon) attached to a manual rotatory joint with 1 m patch cables (Plexbright High Performance, Plexon). Power at the end of the patch cables was measured before each experiment with a portable optical power meter (Thor Labs) and stimulation trains were

generated using V2.2 Radiant software (Plexon). Animals with mistargeted viral injections or optic fibers were excluded from the analysis.

## Calcium endoscopy

Following GCaMP6 virus infection, GRIN lens imaging canulae (Model L, Doric Lenses) were stereo-taxically implanted over the VMHvl and mice allowed to recover for 2–3 weeks followed by 2 weeks isolation under reverse light-cycle and at least 3 days habituation to the microscope plugging and unplugging procedure. Mice were then habituated to the behavioral apparatus for 3 days, and for the second and third days the microscope body was attached. The following week mice (N = 4) were subjected to two consecutive social defeats tests (day 1: social defeat, day 2: memory) over 4 days, and 2–3 days later, three consecutive aggression tests over 3 days. In a second group of mice (N = 3), the testing order as reversed. Esr1 mice (N = 3) were subjected to two consecutive social defeats tests (day 1: social defeat, day 2: memory) over 4 days. Recordings were done with 15–25% LED intensity using 50 ms or 100 ms exposure times. Calcium imaging data was successfully collected from 19/32 animals that underwent surgery. Animals with mistargeted endoscope placements, very few (<15) recorded neurons, recordings exhibiting excessive movement artifacts, or who showed insufficient instances of relevant behaviors were excluded from the analysis (9/19).

## Calcium imaging analysis

Image processing was done using Fiji software. Briefly, videos were first loaded as a stack of images in. tiff format. The stack was duplicated and for each frame a background frame was generated using a band pass filter (lower band: 100, higher band: 10,000). Next, each frame of the recording was divided by the corresponding background frame and the resulting background-filtered stack was aligned using TurboReg via the translation batch algorithm. Neuronal ROIs were manually selected from the maximum intensity projection image, with detection aided by inspecting recordings at increased speed to discern dim neurons with slow dynamics. Finally, the mean intensity ROI traces were extracted and ΔF/F was calculated where F is the mean intensity over the entire recording period. To track ROIs over different recordings and days, an ROI mask was projected onto each new recording and translated if necessary to account for possible field of view movement between recordings (*Figure 1—figure supplement 1*). ROIs that could not be assigned to the mask were treated as new ROIs.

## Data analysis

For the analysis of neuronal response properties, receiver operating characteristic (ROC) curves were calculated using custom Python scripts (scikit-learn) (*Krzywkowski and Penna, 2020*). Briefly, all frames in a selected calcium imaging recording were scored as either positive of negative for a particular behavior and a ROC curve was generated for each neuron by plotting the true positive and false positive rates across the distribution. The area under the ROC curve (auROC) was calculated for each recording and averaged across days to give a measure of the responsiveness of a neuron to a given behavior. Neurons with auROC values greater than 0.65 or less than 0.35 were considered to respond to the behavior. Significance of neuronal response auROC values was estimated by calculating the mean and standard deviation (SD) of the distribution of auROC values obtained by shuffling true and false positive labels 1000 times. Neuronal response auROC values $\geq$ 3 SD away from the mean were considered significant. Neuron needed to fulfill both of the above criteria to be classified as positive or negative responding during any given behavior (e.g. Assessment+, Defeat-). For the analysis of auROC values for flight behavior 2 s before and after the initiation of each flight were labeled as true negative and true positive, respectively. Only recordings with at least two instances of relevant behavior were selected for auROC analysis. For example, during pre-defeat context recordings only four animals out of seven transitioned between all the compartments of the apparatus two or more times and the remaining three animals were excluded from analysis. Heat maps were generated by extracting calcium imaging data (meaned z-score over all trials) for individual neurons from before and after the onset or offset of a given behavior. Significant change across behavioral onset or offset was estimated for each neuron by performing a Wilcoxon signed rank-sum test (Python, scikit-learn) on the distribution of values that resulted from averaging the z-scores of frames from the before and after periods for each trial. For linear discriminant analysis (LDA

algorithm, Python, scikit-learn), data was normalized within days by calculating z-scores for each recording separately. Each frame was then expressed as a vector containing calcium values for all neurons. The distance between clusters was quantified by calculating the average distance between data points of a given cluster and all other data points from other clusters. For Venn diagrams, we included only neurons responding to the behaviors (e.g. Defeat+) that were recorded across all relevant behaviors, resulting in different numbers of neurons across Venn analyses (*Supplementary file 1*). Probability of chance overlap was computed by multiplying the probabilities of responsive neurons. Significance was assessed using Fisher's test.

### Statistical analysis

Prism Graphpad five software or custom scripts in Python were used to generate graphs and perform statistical analysis. For calcium imaging Wilcoxon rank-sum test (Python, scikit-learn package) was performed. For cFos experiments, one-way ANOVA with Tukey's post-hoc test was used. For optogenetic experiments repeated measures ANOVA with Tukey's post-hoc test was used. For assessing significance of overlap between different populations of neurons Fisher's test was used. For assessing significance differences in LDA representations the one-way ANOVA and T test were used.

### Data and code availability

Custom code written for this study is made available on GitHub platform https://github.com/GrossLab/VMHvl-Calcium-Imaging-and-Visualisation (copy archived at https://github.com/elifesciences-publications/VMHvl-Calcium-Imaging-and-Visualisation). Behavioral and imaging data will be made available upon reasonable request.

## Acknowledgements

We thank Dominic Evans and Tiago Branco for sharing expertise in calcium endoscopy and Daniel Rossier, Irene Ayuso, Senthil Deivasigamani, Hiroki Asari and Santiago Rompani for helpful comments on the manuscript. The work was supported by EMBL and the European Research Council (ERC) Advanced Grant COREFEAR to CTG.

## Additional information

### Funding

| Funder | Grant reference number | Author |
| --- | --- | --- |
| H2020 European Research Council | AdG COREFEAR | Piotr Krzywkowski<br>Cornelius T Gross |

The funders had no role in study design, data collection and interpretation, or the decision to submit the work for publication.

### Author contributions

Piotr Krzywkowski, Conceptualization, Data curation, Software, Formal analysis, Validation, Investigation, Visualization, Methodology, Writing - review and editing; Beatrice Penna, Software, Visualization; Cornelius T Gross, Conceptualization, Resources, Supervision, Funding acquisition, Methodology, Writing - original draft, Project administration, Writing - review and editing

### Author ORCIDs

Piotr Krzywkowski (iD) https://orcid.org/0000-0003-4004-1678
Cornelius T Gross (iD) https://orcid.org/0000-0001-9129-1322

### Ethics

Animal experimentation: All experimental procedures involving the use of animals were carried out in accordance with EU Directive 2010/63/EU and under approval of the EMBL Animal Use Committee and Italian Ministry of Health License 541/2015-PR to CG.

### Decision letter and Author response
Decision letter https://doi.org/10.7554/eLife.57148.sa1
Author response https://doi.org/10.7554/eLife.57148.sa2

---

## Additional files

### Supplementary files
• Supplementary file 1. Table showing number of ROIs for each mouse that were assessed for the indicated behavior. Territory includes both Home and Far chamber cells.

• Transparent reporting form

### Data availability
All data generated or analysed during this study are either included in the manuscript and supporting files or deposited to Dryad (DOI: 10.5061/dryad.7pvmcvdrb). Dataset deposited allows for full analysis, although raw endoscope recording files in .tiff format and video camera recordings from which the raw data (contained in the dataset) was extracted were not deposited due to the size in range of Terabytes. To request access to raw .tiff format recordings please contact Dr Cornelius Gross. Code used to analyse the data in the study is deposited on GitHub.

The following dataset was generated:

| Author(s) | Year | Dataset title | Dataset URL | Database and Identifier |
|---|---|---|---|---|
| Piotr Krzywkowski | 2020 | Data from: Dynamic encoding of social threat and spatial context in the hypothalamus - Calcium activity recordings and behavioural dataset | https://doi.org/10.5061/dryad.7pvmcvdrb | 10.5061/dryad.7pvmcvdrb, 10.5061/dryad.7pvmcvdrb |

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
