## [Decision Letter]

**Acceptance summary:**

This study includes a particularly intriguing use of calcium imaging to monitor single neuron activity deep in the brain during behavior to demonstrate that ventromedial hypothalamic (VMH) cells respond during socially threatening experiences. The most compelling aspect of the results was the finding that VMH cells come to encode contextual features of the defeat environment or respond in the safety of the home cage following social defeat experiences. These findings show how aversive social experiences reconfigure hypothalamic circuits to drive learned adaptive behaviors.

**Decision letter after peer review:**

Thank you for submitting your article "Dynamic encoding of social threat and spatial context in the hypothalamus" for consideration by *eLife*. Your article has been reviewed by three peer reviewers and the evaluation has been overseen by a Reviewing Editor and Kate Wassum as the Senior Editor. The reviewers have opted to remain anonymous.

The reviewers have discussed the reviews with one another and the Reviewing Editor has drafted this decision to help you prepare a revised submission.

As the editors have judged that your manuscript is of interest, but as described below that additional analyses and data are required before a final decision can be reached, we would like to draw your attention to changes in our revision policy that we have made in response to COVID-19 (https://elifesciences.org/articles/57162). First, because many researchers have temporarily lost access to the labs, we will give authors as much time as they need to submit revised manuscripts. We are also offering, if you choose, to post the manuscript to bioRxiv (if it is not already there) along with this decision letter and a formal designation that the manuscript is 'in revision at *eLife*'. Please let us know if you would like to pursue this option. (If your work is more suitable for medRxiv, you will need to post the preprint yourself, as the mechanisms for us to do so are still in development.)

Summary:

In this manuscript, using newly designed behavioral paradigms and microendoscopic calcium imaging of deep brain area, Krzywkowski et al. reported that past social experience reshaped activity patterns of social threat and spatial context neurons in the VMHvl. They first designed a new behavioral paradigm for performing rodent social defeat in far chamber, resident-intruder test in home chamber, and approach-avoidance behaviors without manual interference. Next, they applied in vivo microendoscopy to image calcium response of VMHvl neurons at single-cell level in free-moving mice subjected to social defeat. Using these approaches the authors revealed that neurons in the ventrolateral subregion of the VMH (VMHvl) are activated by social threat as well as the context in which that social threat occurred. Furthermore, using the resident-intruder test in home chamber, the authors found that the VMHvl neurons showed an overlapping encoding of social defense and aggression. Moreover, the authors found that social defeat experience altered neurons in the VMHvl such that subsequent optogenetic activation of VMHvl could elicit avoidance of the defeat context. Collectively, these findings reveal a novel, dynamic role for the VMHvl in mediating both social threat and the contextual factors which control these threats.

Overall, this interesting set of data represents a conceptual advance in understanding the role of the VMHvl in aggressive behavior and contextual coding and the reviewers are generally enthusiastic about the work. However, there are a number of important analytic concerns and key behavioral measures are missing. In addition, tracking single cells across days during calcium imaging experiments is not trivial and the authors need a more rigorous assessment of this. Though no new experiments are required, the addition of new analyses, existing data and textual revisions is therefore necessary before a decision can be reached.

Essential revisions:

1) The authors wrote that "Many neurons (100/246, 40%, Social+) showed an increase in activity during close social interaction that returned to baseline levels during the subsequent approach-avoidance phase". It is unclear whether "close social interaction" means the whole period in the far chamber or every social interaction epoch with the aggressor. Please clarify. In Figure 1F, G, H, the authors only showed overall calcium activities in and out of the far chamber. It should be possible to align the signal to every social interaction epoch as shown in Figure 1I, J, K, which will provide more information on the timing of Ca signals. In addition, since the VMHvl also responded to contexts as shown in Figure 2, a control experiment should be performed for Figure 1L, with no aggressor in the far chamber to exclude the context-coding neurons from Social+/- neurons.

2) The definition of Flight- neurons and calculation of their auROC are confusing. An auROC value smaller than 0.5 means that the neuron has a higher tendency to fire not during the event. However, in the bottom trace of Figure 1O, the activity of the example Flight- neuron is elevated right before the flight behavior and declined during the flight, but still above the baseline. According to this pattern, its auROC value should be bigger than 0.5, contradicting to its definition (auROC < 0.35) as indicated in Figure 1Q.

3) The authors concluded that "defeat has a unique capacity to alter the encoding of social cues in VMHvl. Notably, this change was found only when defeats preceded, but not when they followed aggression experiences". Interesting result, but it is not clear whether this difference in VMHvl neural coding is a consequence or a cause of different behavioral performance owing to different sequential order of defeat and aggression. The authors seem to implicate the behaviors are the same with the statement that "The experimental animals showed similar frequencies of behaviors across the two days of social defeat (Figure 4—figure supplement 1A)". However, they mixed behavioral results of all mice tested with different sequential orders. Please separate the two groups with defeat before or after aggression and quantify the behaviors again.

4) For Figure 1, no behavioral data is shown or described at all in the main text or supplement in relation to Figure 1. Do all the mice experience defeat? Is the degree to which a mouse is "defeated" (i.e. number of attacks, etc.), related at all to neuronal activity patterns? How much time on average do mice spend avoiding or approaching the defeater? The authors should display and describe their behavior data for this initial Results section.

5) In the subsection “Encoding of social threat”, Social- neurons are described as showing either no change during defeat/social investigation and an increase in activity during the approach/avoidance phase or a reduction during the defeat phase coupled with an increase in the approach/avoidance phase. However, the data for these two options is pooled under the "Social-" category, where really, they should be pooled under an Avoidance+ category. This pooling seems to ignore the potential differences between the 3 types of neurons the authors identify: 1) Social+/Avoidance- cells, 2) Social no change/Avoidance+ cells, and 3) Social-/Avoidance+ cells. The reviewer understands that separating the data further would increase the complexity of these data and introduce additional factors to consider, however, perhaps a supplementary figure breaking down these groups could be included, or at the least, a revision to the text explaining why neuron classes 2 and 3 are being merged.

6) The authors say that they use a cutoff of >0.65 and <0.35 to classify cells as being responsive to a given behavior, but also say they use a shuffling procedure and a significance cutoff of >/= 3SD from the mean. It is not clear when these different criteria were used. For example, this leads to some confusion when examining Figure 1. For Figure 1H, K, N and Q frequency histograms, there are colored portions of lines below 0.65 and above 0.35. Are the cells counted here assessed using the 3 SD criterion? Relatedly, for cell counts in the Venn diagram in Figure 1T, which criterion was used to classify them? This same confusion applies to all imaging figures.

7) Similar to point 4, the authors show no behavioral data for Figure 2. How much time do mice spend in each of the chambers during the test phase? Is it related to the degree to which they were defeated the prior day? These should be data that the authors already have and can put figures together for without running any extra experiments.

8) Figure 3A-D. There is usually a small percentage of resident animals in a resident intruder assay that fail to exhibit any fighting (as seen in papers cited by the authors throughout the text). Did the authors see aggression in all of their residents? This is important as it speaks to the degree of overlap observed in this experiment (for instance, perhaps the degree of overlap seen is due to the animal(s) that didn't fight?).

9) The authors wrote "Defeat+ and Assessment+ cells showed a high degree of overlap (32% vs. chance 20%, P > 0.1)". How can the conclusion of overlap be made based on a non-significant p value?

10) The cluster distance statistical measure/comparison used for the contextual coding and effects of repeated defeat on population activity should also be used for the analysis of the effects of repeated defeat on ESR1+ cells (related to data in Figure 4K-L).

11) The authors concluded that "VMHvl neurons promote aggression and defense" based on data from Figure 5. Yet only defensive behavior, no aggression, was analyzed in Figure 5.

12) Is the "social interaction" behavior in Figure 5 the same as "close social interaction" defined in Figure 1? If yes, please explain the discrepancy between optogenetic manipulation (activation of VMHvl Esr1+ neurons decreased social interaction, Figure 5) and endoscope recording (more Social+ neurons than Social- neurons found in VMHvl, Figure 1).

13) Are the mice used for the data presented in Figure 5E the same as 5D or are these separate cohorts of mice? If the former, could the repeated optogenetic activation explain some of the effects observed? Please explain/clarify in text.

14) Given that the endoscopic imaging lasted for ~1 week (as shown in Figure 1—figure supplement 1A-C) and several comparisons were conducted across days, it is critically important to verify the stability of imaging across different recording days. Please show maps of spatial filters of all cells from each day of imaging and the overlaid filters (see Extended Data Figure 7 in Remedios et al.'s paper (Remedios et al., 2017) for reference). In addition, the authors should think about using a more quantitative measure of ROI stability potentially using an existing algorithm for aligning ROIs across days (such as Sheintuch, L….Ziv, Y. Cell Reports 21(4) pg. 1102, 2017).

15) In the optogenetic experiments the authors say they use a YFP alone control (subsection “Functional remodeling of Esr1+ neurons by social defeat”), but don't show the data. This data should be shown.

16) Please ensure you include full statistical reporting including F, t statistic, degrees of freedom, exact p value, etc.

---

## [Author Response]

Essential revisions:1) The authors wrote that "Many neurons (100/246, 40%, Social+) showed an increase in activity during close social interaction that returned to baseline levels during the subsequent approach-avoidance phase". It is unclear whether "close social interaction" means the whole period in the far chamber or every social interaction epoch with the aggressor. Please clarify.

Social+ cells are ROIs that respectively increased or decreased their signal during the entire period from the introduction of the aggressor into the far chamber to the moment the mouse was allowed to escape. We agree that this may have been confusing and have now replaced the term “close social interaction” in this context with “during the social defeat phase”.

In Figure 1F, G, H, the authors only showed overall calcium activities in and out of the far chamber. It should be possible to align the signal to every social interaction epoch as shown in Figure 1I, J, K, which will provide more information on the timing of Ca signals.

We chose to designate these neurons Social+ precisely because they show robust tonic activity while the animal is in the defeat chamber. We note that this activity is very similar to the activity observed in previous studies during social interaction (Remedios et al., 2017). The reviewer is right to point out that there is an additional phasic component to the Social+ activity. We have pulled out the strongest of these components that correlated with the epochs in which the animal was chased and bitten by the aggressor, and called these Defeat+ neurons. Notice that Defeat+ and Social+ neurons strongly overlapped (Figure 2—figure supplement 1), demonstrating a simultaneous tonic and phasic modulation of some neurons in VMHvl.

In addition, since the VMHvl also responded to contexts as shown in Figure 2, a control experiment should be performed for Figure 1L, with no aggressor in the far chamber to exclude the context-coding neurons from Social+/- neurons.

The requested control experiment is shown in (Figure 2M, N, R, T) in which we recorded neural activity during exploration of the apparatus prior to social interaction/defeat and found only very weak encoding of the far chamber. These data suggest that Social+ neurons were not responding to the context, but rather to the presence of the conspecific. We note that there was a significant overlap between Social+ and Far chamber+ neurons (Figure 2—figure supplement 1A), showing that neurons that later responded when the animal was in the defeat chamber with the aggressor (Social+) were not yet active when in the far chamber before exposure to the aggressor, but only thereafter (Far chamber+).

2) The definition of Flight- neurons and calculation of their auROC are confusing. An auROC value smaller than 0.5 means that the neuron has a higher tendency to fire not during the event. However, in the bottom trace of Figure 1O, the activity of the example Flight- neuron is elevated right before the flight behavior and declined during the flight, but still above the baseline. According to this pattern, its auROC value should be bigger than 0.5, contradicting to its definition (auROC < 0.35) as indicated in Figure 1Q.

As noted in the Materials and methods section, due to the short duration of flights we calculated auROCs for flight using 2 seconds before and 2 seconds after its initiation. Thus, Flight+ and Flight- describe the response of the neurons at the decision point of flight, separating those neurons whose activity went down from those that went up. We note that Flight- cells highly overlap with Assessment+ cells suggesting that there may be just two major classes of responses to flight, one going up and the other going down. Other studies in VMH have found cells with similar flight response patterns (Masferrer et al., 2018).

3) The authors concluded that "defeat has a unique capacity to alter the encoding of social cues in VMHvl. Notably, this change was found only when defeats preceded, but not when they followed aggression experiences". Interesting result, but it is not clear whether this difference in VMHvl neural coding is a consequence or a cause of different behavioral performance owing to different sequential order of defeat and aggression. The authors seem to implicate the behaviors are the same with the statement that "The experimental animals showed similar frequencies of behaviors across the two days of social defeat (Figure 4—figure supplement 1A)". However, they mixed behavioral results of all mice tested with different sequential orders. Please separate the two groups with defeat before or after aggression and quantify the behaviors again.

We agree that differences in behaviour between the forward and reverse groups could have influenced our ensemble findings and have now analysed the data separately (Figure 4—figure supplement 1B-D). Both forward and reverse groups were repeatedly attacked and show clear defensive responses (defence, upright, assessment) during defeat. However, there was a trend for the reverse group (agg-agg-agg-def-def) to show fewer defensive responses and this reached significance for upright postures. Such a difference is consistent with animals that have experienced aggression being somewhat less susceptible to defeat and leaves open the possibility that some of the difference in the impact of defeat on ensemble coding in the forward and reversed groups depend on differences in behavioral performance. However, the significant difference in ensemble encoding between defeat day 1 and defeat day 2 seen in the forward, but not reversed group was not associated with a difference in defensive behavior (Figure 4—figure supplement 1B-D). This lack of correlation suggests that ensemble encoding differences do not depend on differences in behaviour. We have now expanded the text to reflect this argument.

4) For Figure 1, no behavioral data is shown or described at all in the main text or supplement in relation to Figure 1. Do all the mice experience defeat? Is the degree to which a mouse is "defeated" (i.e. number of attacks, etc.), related at all to neuronal activity patterns? How much time on average do mice spend avoiding or approaching the defeater? The authors should display and describe their behavior data for this initial Results section.

Yes – all mice experienced defeat. Figure 4—figure supplement 1 presents a quantification of defeat and upright behavior for each mouse used in Figure 1 across defeat days. To extend the analysis we have now added quantifications of assessment behavior during the post-defeat approach-avoidance phase (Figure 4—figure supplement 1). No patterns emerged between defense or upright behaviors and assessment (Figure 4—figure supplement 1), although we acknowledge that this analysis was highly underpowered due to the relatively small number of animals used in the imaging studies.

5) In the subsection “Encoding of social threat”, Social- neurons are described as showing either no change during defeat/social investigation and an increase in activity during the approach/avoidance phase or a reduction during the defeat phase coupled with an increase in the approach/avoidance phase. However, the data for these two options is pooled under the "Social-" category, where really, they should be pooled under an Avoidance+ category. This pooling seems to ignore the potential differences between the 3 types of neurons the authors identify: 1) Social+/Avoidance- cells, 2) Social no change/Avoidance+ cells, and 3) Social-/Avoidance+ cells. The reviewer understands that separating the data further would increase the complexity of these data and introduce additional factors to consider, however, perhaps a supplementary figure breaking down these groups could be included, or at the least, a revision to the text explaining why neuron classes 2 and 3 are being merged.

We agree with the reviewer that a distinction could potentially be made between cells that show either no change or a decrease from baseline during the defeat/social phase. However, several technical issues made it difficult for us to definitively segregate these cell types. First, for some animals video recording was started only shortly before the intruder was placed into the chamber and we could not be sure we had properly captured baseline activity. The relatively short baseline periods made it difficult to obtain the significant auROC values needed to distinguish these cells. Second, because confident identification of Social cells involved assessment across 2 days and baseline activity on day 2 could be influenced by day 1, we did not feel confident in drawing any conclusions about a separate identity of these cell types and combined them. New experiments would have to be carried out, potentially with in vivo electrophysiology, aimed at explicitly testing this hypothesis, although we note that Social+ like cells have been reported in the past (Remedios et al., 2017). We have now added a short statement to the text to indicate why the two putative classes were combined: “Although it appeared that Social- neurons might represent two distinct populations, one that decreased and another that remained unchanged during the social defeat phase, it was not possible to statistically distinguish them.”

6) The authors say that they use a cutoff of >0.65 and <0.35 to classify cells as being responsive to a given behavior, but also say they use a shuffling procedure and a significance cutoff of >/= 3SD from the mean. It is not clear when these different criteria were used. For example, this leads to some confusion when examining Figure 1. For Figure 1H, K, N and Q frequency histograms, there are colored portions of lines below 0.65 and above 0.35. Are the cells counted here assessed using the 3 SD criterion? Relatedly, for cell counts in the Venn diagram in Figure 1T, which criterion was used to classify them? This same confusion applies to all imaging figures.

We apologize for inadvertently failing to mention that each neuron had to fulfil both criteria (cutoff >0.65/<0.35 and 3SD from mean) to be classified as +/-. As a result, the coloured section of the histogram (Figure 1H, K, N, Q), between 0.35 and 0.65 was not counted as +/- neurons for any of the analyses, including the Venn diagrams. We have now added a clarifying statement in the Materials and methods section.

7) Similar to point 4, the authors show no behavioral data for Figure 2. How much time do mice spend in each of the chambers during the test phase? Is it related to the degree to which they were defeated the prior day? These should be data that the authors already have and can put figures together for without running any extra experiments.

We now include data showing time spent in the far, corridor and home chambers for each day and group (Figure 2—figure supplement 3A-C). No correlation between time spent in any chamber and defeat duration was found (Figure 2—figure supplement 3C).

8) Figure 3A-D. There is usually a small percentage of resident animals in a resident intruder assay that fail to exhibit any fighting (as seen in papers cited by the authors throughout the text). Did the authors see aggression in all of their residents? This is important as it speaks to the degree of overlap observed in this experiment (for instance, perhaps the degree of overlap seen is due to the animal(s) that didn't fight?).

Indeed, two animals failed to exhibit attacks in the resident-intruder test (see Supplementary file 1). However, our overlap calculations were restricted to neurons that were recorded during all relevant behaviors. For example, no neurons were included in the analysis for Attack+ neurons from animals showing no attacks. This is stated in the Materials and methods section.

9) The authors wrote "Defeat+ and Assessment+ cells showed a high degree of overlap (32% vs. chance 20%, P > 0.1)". How can the conclusion of overlap be made based on a non-significant p value?

We are sorry if we implied that the overlap was significant. The two cell types showed a 32% overlap, which however, did not reach significance when compared to chance. We have now changed the phrase “…high…” to “…moderate, but not significant…” to explicitly indicate that the overlap did not reach significance.

10) The cluster distance statistical measure/comparison used for the contextual coding and effects of repeated defeat on population activity should also be used for the analysis of the effects of repeated defeat on ESR1+ cells (related to data in Figure 4K-L).

Unfortunately, for technical reasons we don’t have the data for contextual exploration from before the defeat for this group of animals. However, we have now added the mean cluster distance after defeat for contextual coding for Esr1+ cells (Figure 2—figure supplement 2D) which show a similar pattern to that derived from the non-cell-type specific recordings (Figure 2Q). For the analysis of Esr1+ cells ensemble changes following repeated defeat we cannot repeat exactly the same analysis as we have only a single inter-cluster distance between defeat 1 and 2 and cannot compare it to any other inter-cluster distances like we did for the non-cell-type specific recordings where each animal underwent 3 aggression experiences in addition to the defeats. As a result, we performed direct statistical comparison of the defeats (Kolmogorov-Smirnov test P = 0.0001 for each mouse) as well as a machine learning based comparison of predictive accuracy using Esr1+ and non-cell-type specific data (Figure 4—figure supplement 3D). We also compared inner-cluster distance (distance between all pairs of points within a cluster) for defeat 1 and defeat 2 with inter-cluster distance (distance between all pairs of points between clusters) of defeat 1 and 2 Figure 4—figure supplement 3E).

11) The authors concluded that "VMHvl neurons promote aggression and defense" based on data from Figure 5. Yet only defensive behavior, no aggression, was analyzed in Figure 5.

We were referring to our data on defense and those published by others showing that ChR2 stimulation of Esr1+ cells can elicit attacks against certain opponents (e.g.

castrated males, females). However, to avoid confusion we have now removed the statement.

12) Is the "social interaction" behavior in Figure 5 the same as "close social interaction" defined in Figure 1? If yes, please explain the discrepancy between optogenetic manipulation (activation of VMHvl Esr1+ neurons decreased social interaction, Figure 5) and endoscope recording (more Social+ neurons than Social- neurons found in VMHvl, Figure 1).

We apologize if our nomenclature here was confusing. “social interaction” (e.g. Figure 5) occurred when the mice were performing defeat, upright, orientating, sniffing, or following behaviors (see Materials and methods), while “close social interaction” (e.g. Figure 1) refers to the entire time when the mice where in the defeat chamber. We have now clarified this distinction by making changes to the text to explicitly define these terms and use them consistently across the manuscript: “Close social interaction” is now referred to as “social defeat phase”, and “social interaction” has been retained.

13) Are the mice used for the data presented in Figure 5E the same as 5D or are these separate cohorts of mice? If the former, could the repeated optogenetic activation explain some of the effects observed? Please explain/clarify in text.

These are the same groups of mice with the addition of three mice that served as controls for repeated optogenetic activation and were not stimulated before defeat. Accordingly, in Figure 5E there are four blue traces while in Figure 5D there are seven that include the three control animals. Comparison of these mice shows that there was no apparent impact of prior stimulation. We now explicitly mention this in the Materials and methods section.

14) Given that the endoscopic imaging lasted for ~1 week (as shown in Figure 1—figure supplement 1A-C) and several comparisons were conducted across days, it is critically important to verify the stability of imaging across different recording days. Please show maps of spatial filters of all cells from each day of imaging and the overlaid filters (see Extended Data Figure 7 in Remedios et al.'s paper (Remedios et al., 2017) for reference). In addition, the authors should think about using a more quantitative measure of ROI stability potentially using an existing algorithm for aligning ROIs across days (such as Sheintuch, L….Ziv, Y. Cell Reports 21(4) pg. 1102, 2017).

We have now added spatial filter maps for each animal and day including a day-by-day comparison series for a representative animal (Figure 1—figure supplement 2). We appreciate that algorithms measuring ROI stability, such as in Sheintuch et al., 2017, can be useful for studies depending on automatic unit identification to image hundreds to thousands of neurons at once. However, the relatively small number of units in our study (30-60 neurons per animal) could be assessed reliably using manual identification following previously published methods (e.g. Evans et al., Nature, 2018). In addition, it is difficult in our opinion to evaluate which method is better as we are not aware of any study comparing manual approaches to such algorithms (centroid distance, spatial correlation). We also note that studies using single photon widefield miniscopes for single unit GCaMP imaging in deep brain nuclei always seem to use semi-automated methods where units are manually inspected and selected, possibly due to the poor performance of automated algorithms on the low contrast signal-to-noise images involved (Jennings et al., Cell, 2015; Remedios et al., 2017).

15) In the optogenetic experiments the authors say they use a YFP alone control (subsection “Functional remodeling of Esr1+ neurons by social defeat”), but don't show the data. This data should be shown.

We apologize if the figure legend was not clear. Data from the YFP control mice is present as grey lines (Figure 5D, E). We have now adjusted the figure legend to make this clearer.

16) Please ensure you include full statistical reporting including F, t statistic, degrees of freedom, exact p value, etc.

We have now added full statistical reporting for all figures.